# Narratives and counter-narratives in religious responses to COVID-19: A computational text analysis

**Ellen Idler**[1]*, **John A. Bernau**[2], **Dimitrios Zaras**[1]

**1** Department of Sociology, Emory University, Atlanta, Georgia, United States of America, **2** Center for the Study of Law and Religion, Emory University, Atlanta, Georgia, United States of America

☯ These authors contributed equally to this work.
* eidler@emory.edu

**Data Availability Statement:** We used publicly-available data from the New York Times and included our search terms and exclusions in the manuscript. Links to documents published by the CDC, WHO, and faith-based organizations are included in the supplementary files.

## Abstract

Religious responses to COVID-19 as portrayed in a major news source raise the issue of conflict or cooperation between religious bodies and public health authorities. We compared articles in the *New York Times* relating to religion and COVID-19 with the COVID-19 statements posted on 63 faith-based organizations' web sites, and with the guidance documents published by the Centers for Disease Control and Prevention (CDC) and World Health Organization (WHO) specifically for religious bodies. We used computational text analysis to identify and compare sentiments and topics in the three bodies of text. Sentiment analysis showed consistent positive values for faith-based organizations' texts throughout the period. The initial negative sentiment of religion—COVID-19 coverage in the *New York Times* rose over the period and eventually converged with the consistently positive sentiment of faith-based documents. In our topic modelling analysis, rank order and regression analysis showed that topic prevalence was similar in the faith-based and public health sources, and both showed statistically significant differences from the *New York Times*. We conclude that there is evidence of both narratives and counter-narratives, and that these showed demonstrable shifts over time. Text analysis of public documents shows alignment of the interests of public health and religious bodies, which can be discerned for the benefit of communities if parties are trusted and religious messages are consistent with public health communications.

## Introduction

Religious institutions have been much in the news since the start of the COVID-19 pandemic. Some very early outbreaks were traced to religious groups. In South Korea [1] the Shincheonji Church of Jesus in Daegu was the source of an outbreak in late February that infected hundreds of church members. In Washington State [2], after a lengthy choir rehearsal at a church, 45 singers became ill with the coronavirus and two died. In rural Arkansas in mid-March, a pastor and his wife tested positive; 92 persons had been exposed to them; 38% were confirmed

**Funding:** A modest amount of internal funding was provided by the Department of Sociology, Emory University. DZ received the award. The Department of Sociology played no role in study design, data collection and analysis, or preparation of the manuscript.

**Competing interests:** The authors have declared that no competing interests exist.

COVID-19 cases and 3 persons died [3]. Large social gatherings with unison speaking and/or singing—as often occurs with religious services—are important potential exposures to airborne infection.

A second appearance of religious groups in news about the pandemic stems from the messages promoted by religious leaders about appropriate responses to the coronavirus. Politically conservative religious voices in the US have made arguments against public health directives limiting religious mass gatherings on the grounds of constitutional protections of religious freedom [4], thereby politicizing the issue. A funeral drawing thousands of mourners in an Orthodox Jewish Community in Brooklyn was broken up by the New York City police [5]. A small but significant minority of Evangelical Christian mega-churches continued to hold large services despite municipal restrictions [6]. On June 1st, 2020, the Supreme Court denied an application by a California church challenging Governor Gavin Newsom's Stay-at-Home order [7], but on November 25th, 2020 with the appointment of a new justice, the Court ruled against New York Governor Andrew Cuomo's restrictions on the size of religious gatherings [8]. These events could lead the non-religious public to form impressions of religious people in general as anti-scientific, and/or politically focused on individual rights and freedoms while showing little concern for the safety of the collective and the protection of others. This message of conflict between religion and public health has been a seemingly dominant narrative in the media during the COVID-19 pandemic to date.

Such high-profile events exemplifying the conflict narrative are at odds with US nationally representative survey data showing that fewer than 10% of Americans, including a majority of Evangelical Christians and Republicans, think that in-person religious services should be permitted without restrictions on size or social distancing [9]. An Associated Press/National Opinion Research Center poll from May 2020 also found that 48% said they had a regular congregation, of which only 7% were said to be currently open, and more respondents of every faith tradition said they were watching services streamed online since the pandemic began [10]. Another poll by the Pew Research Center showed that 79% of US adults thought that houses of worship should be subject to the same restrictions on large gatherings as other organizations [11]. Thus public opinion, including the views of religious Americans, strongly supports public health measures in the current pandemic.

Public actions of religious groups reflect this expressed concern for protecting public health. An interfaith clergy Facebook group with over 7000 members shared new technologies for putting services online, and for performing religious rituals remotely [12]. Some prominent scientists and Christian leaders in the US released "A Christian Statement on Science for Pandemic Times" with more than 7000 signatories supporting alignment of the values of religion and science [13]. A survey of over 200 world religious leaders showed that they had overwhelming support for the actions of their governments to curtail COVID-19, and that they used their influence to encourage adherence to preventive measures in their communities [14]. These views and actions reflect a counter-narrative in which the influence of religious groups on COVID-19 health-related behavior is in line with and promoting of public health recommendations.

Accurate, targeted messaging is essential for public health. Media portrayals of these two opposing narratives can shape the actions of both public health officials and religious group actors. To accurately describe these narratives in the media, we studied articles on religion and COVID-19 in the *New York Times (NYT)*, and compared their characteristics with text obtained from public health authorities and religious and faith-based institutions. Computational text analysis is an important tool for understanding communications and is beginning to appear in public health research [15–18]. One related text analysis of internet searches from 95 countries linked the pandemic to religion by finding an increased number of Google

searches for "prayer" [19], but none have yet examined news media depictions or the response of faith institutions to COVID-19.

In this study we ask, 1) can we identify and describe a dominant narrative of religious disregard for public health, and a counter-narrative of religion's more cooperative role in the pandemic in online text? 2) Do those themes develop over time as the cases and deaths increase? 3) What is the prevalence of topics in the entire body of text, and can differences in the prevalence of topics be seen among the sources? Our computational text analysis provided us with some clear and somewhat surprising answers to these questions.

## Methods

### Data collection

We collected articles published between January 1 and June 30, 2020 in the *New York Times* that contained the following religious and COVID-19 related terms: [COVID*][pandemic] [coronavirus]+[religio*][church][mosque][temple][faith*][worship][clergy][chaplain]. A first sweep yielded 5,465 articles. This sample was reduced to 634 articles after excluding those that contained only one religion term. While the COVID-19 pandemic has extended beyond June 30th, 2020, the goal of our analysis is to analyze discourse in the first six months of the outbreak. Furthermore, while the outbreak was not restricted to New York, it was the site of the largest outbreak in the United States, holding the highest number of confirmed cases until July of 2020 [20]. Thus, *New York Times* was unique in its coverage of both global trends and local spread during this time, and has routinely been used by social scientists [21–24]. The frequency of appearance of these articles is shown in S1 Fig.

To represent public health messaging for religious groups, we collected guidance on COVID-19 issued by the World Health Organization (WHO) and the Centers for Disease Control and Prevention (CDC) specifically for faith communities. These seven documents were released from early April to mid-June and, while smaller in absolute number than the New York Times articles, comprise a near-population of government-issued public health recommendations on the subject; list available in S1 Table.

Our third corpus of texts came from faith-based organizations that issued guidance or posted statements on the pandemic on their web sites. Organizations were identified from the extensive COVID-19 resources materials on the web sites of the Interfaith Health Program (Emory University https://ihpemory.org/) and the Berkley Center (Georgetown University https://berkleycenter.georgetown.edu/topics/COVID-19); and the lists of religious bodies in the United States available at the Association for Religion Data Archives. (https://www.thearda.com/denoms/families/trees/index.asp); and other internet archives. Searching was continued until saturation was reached. The earliest, longest, or most general statement or document was selected, so there is only one statement per group. Only text in English is included. The sample intends to represent the diversity of world faith traditions and faith-inspired aid organizations. These documents were posted from late February to early July. The list of organizations is available in S2 Table.

Our complete corpus of text on religion and COVID-19 consisted of 7 documents from public health sources, 63 documents from faith-based organizations, and 634 documents from the *New York Times*. Together these 704 documents contain 1,014,187 words. The average document length is remarkably consistent across different sources: around 1,400 words. Table 1 provides descriptive statistics for each of the sources in our sample. Previous computational text analyses have demonstrated the utility of these methods when dealing with diverse sets of texts. Nelson et al. [25] use wide search parameters to create a corpus of texts on "inequality" and Baker et al. [26] examine the difference between British broadsheets and

**Table 1. Descriptive statistics for three corpora.**

| Source | N documents | Total vocab size (unique words) | Total word count | Avg word count | SD word count |
|---|---|---|---|---|---|
| Faith-Based Orgs | 63 | 8141 | 87679 | 1398 | 1299 |
| Public Health | 7 | 1901 | 10356 | 1482 | 1175 |
| New York Times | 634 | 38738 | 916152 | 1445 | 807 |
| Total | 704 | 48780 | 1014187 | 1441 | 1093 |

tabloids. Others have explored instant messages, political speeches, social media, legislation, and much more [see 27, 28 for reviews].

## Sentiment analysis

To quantify the affective dimension of texts we calculated "sentiment polarity", using the R package sentimentr [29]. The package 'sentimentr' calculates the sentiment polarity of a text at the sentence level and can aggregate by paragraphs or entire documents. Sentimentr is a dictionary lookup method that augments traditional sentiment dictionary analysis by incorporating weighting for valence shifters. Valence is the positive or negative attitude that is communicated by certain terms. Words that can change the valence of another word or a whole sentence are called valence shifters. Negation, adverbs that increase or decrease intensity (i.e., amplifiers and de-amplifiers), and conjunctions can act as valence shifters, depending on the context [30]. For example, the words "very" or "hardly" could act as valence shifters for a polarized word like "successful". The words in each sentence are searched and compared to a dictionary of polarized words. Positive and negative words are tagged with a +1 and −1 respectively. A context cluster around the polarized word is pulled that defaults to 4 words before and 2 words after that are considered as valence shifters. Each polarized word is then weighted by the function and number of the valence shifters directly surrounding it. Finally, the weighted context clusters are summed and divided by the square root of the word count for each sentence, producing an unbounded sentiment polarity score. To generate a document-level sentiment polarity score we take the mean of all sentence-level sentiment polarity scores. For example, the sentence "I haven't been sad in a long time" would have a sentiment polarity score of 0.18. The sentence "I don't feel so bad after all!" is more positive and would have a sentiment polarity score of 0.28. By contrast, the sentence "Then I'm not happy at all" would have a sentiment polarity score of -0.56 [31]. To show the smoothed trends in sentiment over time, we produced separate LOESS (locally estimated scatterplot smoothing) curves for faith-based organizations and the *New York Times* with document-based sentiment as the dependent variable, and time as a predictor.

## Topic modeling

Topic modeling is an inductive and iterative machine-learning algorithm that identifies groups of words, or "topics", that commonly co-occur in a body of text data to represent semantically interpretable themes [28, 32]. The algorithm, akin to exploratory factor analysis, identifies latent structures in the data—groups of words that co-occur with high frequency, or "topics." These topics reliably match results from qualitative hand-coding and provide an atheoretical and inductive quantitative procedure to evaluate the thematic content of large bodies of text [33]. Typically, this approach yields both expected and unexpected topics, providing both face validity and novel findings. For example, in a topic model of 100 years of newspaper data, one would expect to see topics related to politics, the economy, sports, and international news. However, one might be surprised by the *prevalence* of these topics over time—are international

topics more common as the world becomes interconnected?—or the *content* of these topics over time: how has our language of sports changed? Furthermore, this analysis might also produce subtle or unexpected topics like "nuclear proliferation" or "terrorist threat." In the present study, we employ structural topic modeling (STM) [34] to allow topic prevalence to vary by document source. In other words, we are evaluating how each of our sources give uneven attention to particular topics. This is an important departure from traditional Latent Dirichlet Allocation (LDA) topic modeling, which assumes a single data-generation process and uniform distribution of topics across all documents. In contrast, STM uses document-level metadata— in our case the three sources: public health, faith-based-organizations, or *NYT*—to estimate how the prevalence of topics might vary. This allows a better estimation of diverse corpora [31]. See [30]:3–4 for detailed mathematical formulations.

We tested a range of topic numbers from 10–40, applying goodness of fit tests [32]. The topic model algorithm does not assign substantive labels to these topics. Each author independently assigned labels based on the most frequent words listed in each topic; we discussed these until reaching consensus on a label for the topic.

To examine correlations between topic rankings, we used the Kendall coefficient to test ranking similarity in each possible pair of sources. The Kendall coefficient evaluates the degree of similarity between two sets of ranks given to the same set of objects [35].

In order to compare two ordered sets (on the same set of objects), the approach of Kendall is to count the number of different pairs between these two ordered sets. This number gives a distance between sets called the symmetric difference distance (the symmetric difference is a set operation which associates to two sets the set of elements that belong to only one set).

The symmetric difference distance between two sets of ordered pairs $\mathcal{P}_1$ and $\mathcal{P}_2$ is denoted $d\Delta\,(\mathcal{P}_1, \mathcal{P}_2)$.

Kendall coefficient of correlation is obtained by normalizing the symmetric difference such that it will take values between −1 and +1, with −1 corresponding to the largest possible distance (obtained when one order is the exact reverse of the other order) and +1 corresponding to the smallest possible distance (equal to 0, obtained when both orders are identical). Taking into that the maximum number of pairs which can differ between two sets with

$$\frac{1}{2}N(N-1)$$

elements is equal to

$$N(N-1),$$

this gives the following formula for Kendall rank correlation coefficient:

$$\tau = \frac{\frac{1}{2}N(N-1) - d\Delta\,(\mathcal{P}1,\ \mathcal{P}2)}{\frac{1}{2}N(N-1)} = 1 - \frac{2\,x\,[d\Delta\,(\mathcal{P}1,\ \mathcal{P}2)]}{N(N-1)}$$

Then, to compare the actual (not ranked) prevalence of topics, we computed regression estimates for each topic by source. Using native functions from *stm* package [35], we ran one regression for each topic with topic prevalence as the dependent variable and the three sources as predictors (*NYT* reference group).

As shown in Table 1, the word count and sentence count for each of the corpora are certainly different. However, it is important to note that the two methods we use to analyze the data are independent of the size of the document. Structural topic modeling accounts for differences using document metadata and depends on the relative frequency of the appearance of topics in three bodies of text. Our sentiment analysis takes the sentence as the unit of analysis

and compares the average sentiment score across sentences. Thus both analyses are independent of the length or number of the documents in the three bodies of text.

## Results

The objective of the analysis was to understand the similarities and differences between the affective content (sentiments) and the subject content (topics) of the three text sources, and to identify changes as the COVID-19 pandemic progressed in early 2020.

We begin by comparing the affective content of the *New York Times* and the faith-based organization statements. Fig 1 shows the average sentiment score of articles in the *NYT* and the statements on COVID-19 released by FBOs and religious groups for the period January 1 to early July 2020. Dates of important events, including release of CDC and WHO guidance for religious groups are noted. The mean sentiment score for articles by date is plotted for the *NYT* and the faith-based statements, with a 95% CI. For the faith-based statements the positive sentiment rises and then falls over the period, but remains significantly positive except at the beginning and end where the confidence interval dips slightly below 0.00. The mean sentiment score for *NYT* articles has a straight upward slope; it begins the period negatively, reaches neutrality in mid-April, and remains significantly positive thereafter, overlapping confidence intervals with the faith-based statements. The sentiment scores for the *New York Times* and faith-based organizations suggest dual narratives at the start of the pandemic, but an increasing similarity and ultimate convergence with both ending the period in the range of positive sentiment.

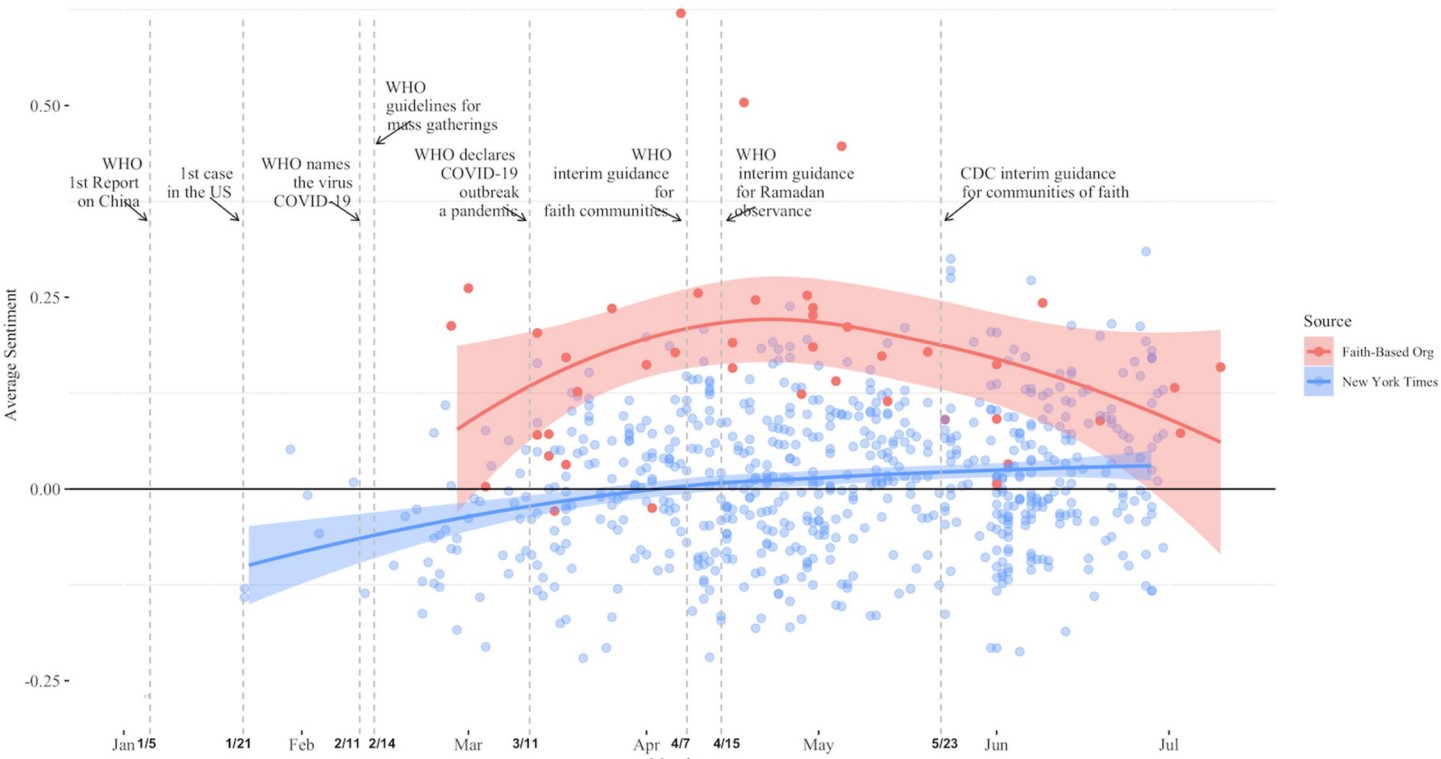

**Fig 1. Average sentiment scores for *New York Times* articles and COVID-19 guidance documents from faith-based organizations and loess curves indicating sentiment trends over time.**

In the next step, to understand differences and similarities in the subject matter, we conducted a topic modeling of the entire corpus of text, combining all three sources. A 30-topic model proved to have the best fit. S3 Table provides the top 30 terms for each of the 30 topics. The sets of terms in most topics were substantively coherent and interesting, pointing to a well-specified model, and the topics could be labelled meaningfully.

To compare content among the three sources, we ranked topics by their proportion in each source, and performed Kendall's rank correlation test. See Table 2. At a glance it is clear that the public health and faith-based sources have roughly similar rankings, with "worship risk considerations" and "religious support" listed first or second for both. By contrast, the top two topics for the *New York Times* were "infection spread" and "presidential campaign". The rank

**Table 2. Ranking of topic prevalence by source: Public health, faith-based groups, *New York Times*.**

| Topic | Public Health ranking | Faith-Based Groups ranking | *New York Times* ranking |
|---|---|---|---|
| **Worship risk considerations** | 1 | 2 | 28 |
| **Religious support** | 2 | 1 | 30 |
| **Muslim holiday observance** | 3 | 9 | 11 |
| **Religious holiday observance** | 4 | 4 | 25 |
| **Christian death** | 5 | 8 | 16 |
| **Global outbreak** | 6 | 10 | 9 |
| **Infection spread** | 7 | 14 | 1 |
| **Government restriction** | 8 | 11 | 10 |
| **Food support** | 9 | 13 | 19 |
| **Care / sickness** | 10 | 7 | 4 |
| **Virtual gathering** | 11 | 3 | 26 |
| **Church outbreak** | 12 | 12 | 20 |
| **Church services** | 13 | 5 | 23 |
| **Political leaders** | 14 | 20 | 8 |
| **Education** | 15 | 19 | 17 |
| **Christian stories** | 16 | 6 | 13 |
| **Jewish funerals** | 17 | 22 | 5 |
| **US government** | 18 | 16 | 17 |
| **New York City** | 19 | 24 | 12 |
| **Brazil** | 20 | 17 | 27 |
| **Weddings** | 21 | 18 | 6 |
| **Home cooking** | 22 | 27 | 24 |
| **Miscellaneous** | 23 | 29 | 18 |
| **Pandemic** | 24 | 23 | 21 |
| **State response** | 25 | 25 | 29 |
| **Sports** | 26 | 26 | 15 |
| **Spirituality** | 27 | 28 | 22 |
| **Black Lives Matter protest** | 28 | 15 | 3 |
| **Presidential campaign** | 29 | 21 | 2 |
| **Arts** | 30 | 30 | 14 |
| Kendall's rank correlation test for similar rankings | Public health v. Faith-based orgs | Public health v. New York Times | *New York Times* v. Faith-based orgs |
| Tau | Tau = 0.6322 | Tau = -0.0942 | Tau = -0.1310 |
| P value | P < 0.0001 | P = 0.4788 | P = 0.3207 |

Note: The characterization of topics shown in column 1 was done independently by the three authors based on the most frequently-occurring words. Any differences were resolved in discussion. The complete lists of words for each of 30 topics are available in S1 Table.

correlation test showed a statistically significant correlation between the topic rankings of the public health and faith-based/religious organizations' documents, or in other words, that these two sources are emphasizing the same topics above others. The tau for those two sources (public health / faith-based) was 0.632 (p<0.001), which indicates a strong level of agreement in their ranked prevalence of the 30 topics. Topic rankings for other pairs of sources did not show statistically significant correlations.

Finally, we sought to quantify the differences in topic prevalence. Table 3 presents the results of regression models for the six topics that showed significant prevalence differences. We ran one model for each topic, with topic prevalence within the entire corpus as the dependent variable, and source as the predictor (*NYT* is the reference category). The six separate regression models are shown in Table 3. Each row depicts the estimated prevalence of a given topic (i.e. Worship Risk) in each corpora (i.e. Faith-Based Orgs or Public Health), with the NYT as reference.

Faith-based groups were more likely than *NYT* to mention "virtual gathering", and less likely to include "presidential campaign", "infection spread", and "Black Lives Matter protest"; in these four cases, public health sources were not different from *NYT*.

Fig 2 depicts these results visually and shows that faith-based and public health documents were significantly more likely than the *NYT* to discuss "worship risk considerations" and "religious support". Insignificant differences are represented by gray circles. For example, Faith-Based Organizations used the "Religious Support" topic more than Public Health documents,

**Table 3. Structural topic modeling regression results.**

| DV: Topic | IV: Source | Topic prevalence estimate (beta) | se | t_value | p_value |
|---|---|---|---|---|---|
| *Worship Risk Considerations* | (Intercept) | 0.016 | 0.006 | 2.948 | 0.003 |
| | *New York Times (ref)* | | | | |
| | Public Health | 0.601 | 0.114 | 5.276 | < 0.001 |
| | Faith-Based Orgs | 0.275 | 0.036 | 7.590 | < 0.001 |
| *Virtual Gathering* | (Intercept) | 0.017 | 0.005 | 3.430 | 0.001 |
| | *New York Times (ref)* | | | | |
| | Public Health | -0.016 | 0.041 | -0.396 | 0.692 |
| | Faith-Based Orgs | 0.043 | 0.020 | 2.151 | 0.032 |
| *Religious Support* | (Intercept) | 0.006 | 0.005 | 1.269 | 0.205 |
| | *New York Times (ref)* | | | | |
| | Public Health | 0.285 | 0.099 | 2.873 | 0.004 |
| | Faith-Based Orgs | 0.382 | 0.029 | 13.069 | < 0.001 |
| *Presidential Campaign* | (Intercept) | 0.056 | 0.008 | 6.684 | < 0.001 |
| | *New York Times (ref)* | | | | |
| | Public Health | -0.055 | 0.068 | -0.812 | 0.417 |
| | Faith-Based Orgs | -0.055 | 0.024 | -2.284 | 0.023 |
| *Infection Spread* | (Intercept) | 0.063 | 0.009 | 7.359 | < 0.001 |
| | *New York Times (ref)* | | | | |
| | Public Health | -0.063 | 0.066 | -0.960 | 0.337 |
| | Faith-Based Orgs | -0.053 | 0.023 | -2.266 | 0.024 |
| *BLM Protest* | (Intercept) | 0.059 | 0.008 | 7.134 | < 0.001 |
| | *New York Times (ref)* | | | | |
| | Public Health | -0.059 | 0.067 | -0.875 | 0.382 |
| | Faith-Based Orgs | -0.055 | 0.024 | -2.240 | 0.025 |

Note: Results from six separate regression models predicting topic prevalence from document source. See native functions from stm package [32].

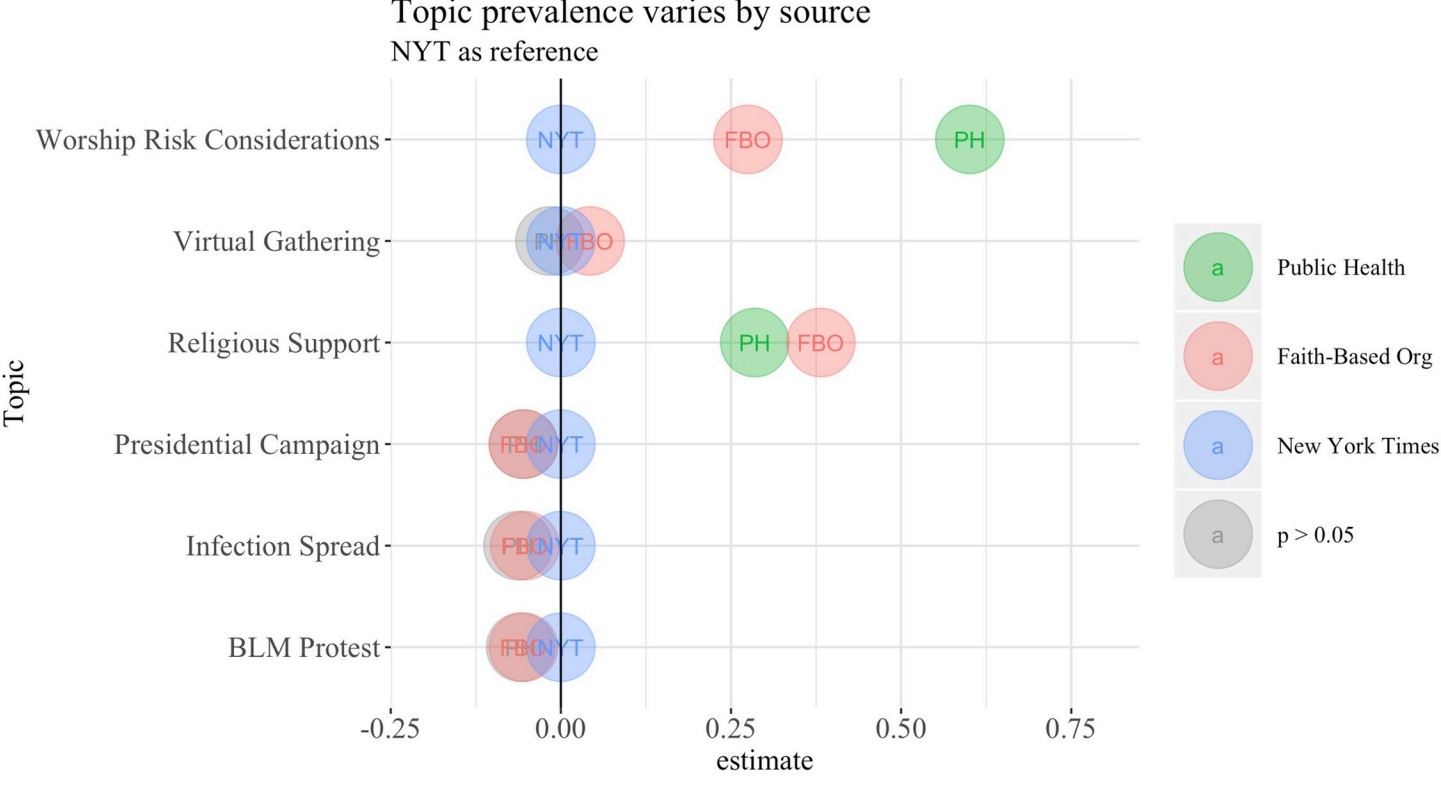

**Fig 2. Topic prevalence differences by source, with *New York Times* as reference group.**

and both employed this topic more than the NYT. Thus, for six of thirty topics, faith-based sources differed from the *New York Times*; in two, public health sources also differed. Importantly, when there were differences, public health documents differed in the same direction as faith-based organizations (more prevalent than in *NYT*).

Overall, there are several findings in our comparison of the portrayals of religious responses to COVID-19 in these three types of source documents. Sentiment analysis showed that from the beginning of the pandemic to early summer, an initially negative sentiment of the *New York Times* trended more positive, and converged with the consistently positive sentiment of the faith-based sources. Our topic modeling of the substantive content of these sources showed that faith-based sources were quite similar to public health sources in their emphases, and that both were different from the *New York Times* in rankings and prevalence of topics.

## Discussion

The aim of this research was to ascertain how the apparent dominant media narrative on religious responses to the COVID-19 pandemic—one in which religious freedom is asserted to reject public health recommendations—compared with the self-representations of faith-based groups. Our assessment of both affect (sentiment) and content (topics) showed—perhaps surprisingly for those attuned to the media's dominant narrative—considerable alignment of faith-based groups with public health recommendations. The efficacy of public health measures depends to a large extent on messaging and communication. Our analysis showed

considerable take-up of CDC and WHO guidelines among religious and faith-based groups, and the propounding of those messages to their members. We would underscore that all three types of documents that we studied were intended to be read by the general public: clergy and laity in religious groups, and readers of national and local news.

In particular, the two topics that were most prevalent for both FBOs and public health were "worship risk considerations" and "religious support". As can be seen in the complete list of topics and their contents in S1 Table, the top-loading words for "worship risk considerations" in order by frequency were: "will-worship-risk-can-gather-church-may-use-congregation-people-distance-community-consider-COVID-health-time-plan-service-member-person-reopen-practice-local-public-hand-guidance-provide-need-social-space". The top-loading words for "religious support" were: "communities-COVID-health-religion-church-support-leader-people-world-faith-can-help-will-respond-need-provide-pandemic-inform-prevent-families-nation-children-also-vulnerable-work-social-resource-crisis-spread-local". Both topics have content that emphasizes the themes of (the risks of) social gatherings, as well as providing for the needs of others, mixed with their strong public health content. Importantly, a core dimension of religion and religious practice is that it is a frequent regular social event for which people congregate. Furthermore, in the large research literature on religion as a predictor of all-cause mortality, it is the social dimension of attendance at religious services that is consistently a protective factor, rather than more solitary, subjective dimensions such as prayer or having a strong religious identity [36, 37]. Thus the historically most health-protective aspect of religious involvement—social gatherings—has ironically become in the pandemic a significant, news-worthy source of deadly COVID-19 spreading events.

There are a number of limitations to our analysis. The *New York Times* as our media source might be considered overly liberal and/or secular, but the paper is routinely used by social scientists for its significant and broad coverage of issues [21, 22]. Moreover, the initial major US outbreak was in New York, giving us early coverage of well-sourced local as well as national reporting. Secondly, the time frame for sampling of all sources ran only through early July, and the situation has evolved significantly since then. A third limitation is that the identification of faith-based and religious groups was restricted to groups with a web presence; therefore small (or even large) independent congregations that did not have a religious hierarchy or regional or national organizational structures would also be less likely to have presented such statements, or to have been found for this study. Finally, a less quantitative, more qualitative study could have done a deeper examination of the articles/statements/guidance documents themselves.

It is essential for public health practice to convey effective messaging to the public, and never has it been both more important and more complex. Religious and faith-based organizations represent an important sector for partnering with public health to disseminate messages and materials, particularly to underserved and hard-to-reach populations. The March 2019 *American Journal of Public Health* Special Section featured accounts of effective partnerships between faith-based organizations and public health at the state, national, and global level [38]. One account with parallels to the present was that of the Ebola crisis, where traditional religious burial practices were initially serious sources of contagion, but where Christian and Muslim groups worked with the WHO to revise "safe burial practices" into "safe and dignified burial practices" that would be more respectful of those religious practices while preventing transmission of the virus [39]. A second account with even more relevance to the present phase of the pandemic concerned the US government's efforts to work with faith-based organizations to promote and deliver H1N1 influenza vaccinations in 2010. A national network of FBOs undertook efforts specific to their communities to overcome economic and cultural barriers to vaccination in their racially and ethnically diverse, but mostly low-income

communities [40]. Such partnerships with "trusted messengers" are continually being touted in COVID-19 vaccine distribution efforts; the models exist, with particular benefit for the hardest-to-reach populations.

## Conclusions

The communication of public health messages to prevent disease and promote health depends for its success on egalitarian partnerships with communities. Such organizational relationships must be respectful, trusting, and willing to find where interests are aligned. Faith-based organizations are important institutions in their local communities—with resources, facilities, leadership, familiarity, and above all the trust of their members and neighbors. Our analysis of public health guidance for faith communities and the self-representations of faith communities' response to COVID-19 finds considerable alignment of language and affect, suggesting a large and important opening to collaboration. There is a considerable distance between the narrative of conflict between religion and science that characterized especially the earliest coverage in the *New York Times*, and the counter-narrative of mutual interest for which we see evidence. This perceived distance should not be a barrier to public health in promoting partnerships with the large majority of faith-based organizations.

## Supporting information

**S1 Table. Public health guidance for religious groups from the centers for disease control and prevention and the world health organization.**
(DOCX)

**S2 Table. Faith-based and religious organization statements and guidance on COVID-19.**
(XLSX)

**S3 Table. Top 30 topic contents with author characterization of themes.**
(XLSX)

**S1 Fig. Frequency of *New York Times* articles by week and category.**
(JPG)

## Acknowledgments

The authors would like to thank Allison Roberts for coding advice.

## Author Contributions

**Conceptualization:** Ellen Idler, John A. Bernau.

**Data curation:** Ellen Idler, John A. Bernau, Dimitrios Zaras.

**Formal analysis:** John A. Bernau, Dimitrios Zaras.

**Investigation:** John A. Bernau.

**Methodology:** John A. Bernau, Dimitrios Zaras.

**Project administration:** Ellen Idler.

**Software:** John A. Bernau, Dimitrios Zaras.

**Supervision:** Ellen Idler, John A. Bernau.

**Validation:** Dimitrios Zaras.

**Visualization:** John A. Bernau, Dimitrios Zaras.

**Writing – original draft:** Ellen Idler, John A. Bernau, Dimitrios Zaras.

**Writing – review & editing:** Ellen Idler, John A. Bernau, Dimitrios Zaras.

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
