## [Decision Letter · Decision Letter 0]

9 Jul 2021

PONE-D-21-08822

Narratives and Counter-Narratives in Religious Responses to Covid-19: A Computational Text Analysis

PLOS ONE

Dear Dr. Idler,

Thank you for submitting your manuscript to PLOS ONE. After careful consideration, we feel that it has merit but does not fully meet PLOS ONE’s publication criteria as it currently stands. Therefore, we invite you to submit a revised version of the manuscript that addresses the points raised during the review process.

We look forward to receiving your revised manuscript.

Kind regards,

Quanquan Gu

Academic Editor

PLOS ONE

Journal Requirements:

2. Please improve statistical reporting and refer to p-values as "p<.001" instead of "p=.000". Our statistical reporting guidelines are available at https://journals.plos.org/plosone/s/submission-guidelines#loc-statistical-reporting.

Reviewers' comments:

Reviewer's Responses to Questions

**Comments to the Author**

1. Is the manuscript technically sound, and do the data support the conclusions?

Reviewer #1: No

Reviewer #2: Partly

2. Has the statistical analysis been performed appropriately and rigorously? 

Reviewer #1: Yes

Reviewer #2: Yes

3. Have the authors made all data underlying the findings in their manuscript fully available?

Reviewer #1: Yes

Reviewer #2: Yes

4. Is the manuscript presented in an intelligible fashion and written in standard English?

Reviewer #1: No

Reviewer #2: Yes

5. Review Comments to the Author

Reviewer #1: 1. In the experiment part, the paper shows a significant correlation between the two sources, and shows an insignificance of the other two correlations. From this observation, the author claims that there's a gap between religion group and science towards the covid pandemic. However, this claim is not sound.

To begin with, the datasets are highly heterogeneous. For instance, guidance from public health is often written by formal language, while media like NYT tends to write in a more informal way. More analysis on the dataset itself in detail should be provided.

More important, the great imbalance with respect to data size must be taken into consideration. With only a few documents from public health and religious group, while thousands of documents from media side, the direct comparison between the two dataset seems not feasible. At least, you should provide some evidences that your comparison can overcome this significant data imbalance.

2. The statistical analysis is straight-forward, and thus appropriate.

3. The authors list the source of all data.

4. The paper is not properly written:

1) In the table 1, what do you mean by p<0.000? p value should be at least a non-negative number.

2) Could you provide a simple explanation of Kendall coefficient, and why you choose this metric instead of the others?

3) A more clear version of figure 3 is needed. You may consider to change to format.

Reviewer #2: This paper studies the religious responses to COVID-19 through the lens of text analysis. The idea seems to be interesting, but the detail of the proposed methods is not very clear. Here are my major concerns:

1. The time period considered in this paper is not long enough. It is well-known that there exists a second wave of COVID-19 after June 30. Therefore, it is also very important to include the analysis after June 30 to support the current claim.

2. The authors may consider including more baseline articles from other news sources besides NYT.

3. The main methods used in this paper are not well explained. The authors should include more details of their proposed sentiment analysis, sentiment score, and topic modeling.

6. PLOS authors have the option to publish the peer review history of their article (what does this mean?). If published, this will include your full peer review and any attached files.

Reviewer #1: No

Reviewer #2: No

---

## [Author Response · Author response to Decision Letter 0]

30 Jul 2021

We have provided detailed responses to the comments of the reviewers and editor in the Response to Reviewers document. We are very grateful for the attention given to our manuscript and the helpful suggestions that were made.

---

## [Decision Letter · Decision Letter 1]

20 Oct 2021

PONE-D-21-08822R1Narratives and Counter-Narratives in Religious Responses to Covid-19: A Computational Text AnalysisPLOS ONE

Dear Dr. Idler,

Thank you for submitting your manuscript to PLOS ONE. After careful consideration, we feel that it has merit but does not fully meet PLOS ONE’s publication criteria as it currently stands. Therefore, we invite you to submit a revised version of the manuscript that addresses the points raised during the review process.

Per Reviewer #2's comment, please add more detailed descriptions, such as some definitions, equations, or even simple examples in the Methods section to help the audience better understand your proposed methods.

We look forward to receiving your revised manuscript.

Kind regards,

Quanquan Gu

Academic Editor

PLOS ONE

Journal Requirements:

Reviewers' comments:

Reviewer's Responses to Questions

**Comments to the Author**

1. If the authors have adequately addressed your comments raised in a previous round of review and you feel that this manuscript is now acceptable for publication, you may indicate that here to bypass the “Comments to the Author” section, enter your conflict of interest statement in the “Confidential to Editor” section, and submit your "Accept" recommendation.

Reviewer #1: All comments have been addressed

Reviewer #2: All comments have been addressed

2. Is the manuscript technically sound, and do the data support the conclusions?

Reviewer #1: Yes

Reviewer #2: Yes

3. Has the statistical analysis been performed appropriately and rigorously? 

Reviewer #1: Yes

Reviewer #2: Yes

4. Have the authors made all data underlying the findings in their manuscript fully available?

Reviewer #1: Yes

Reviewer #2: Yes

5. Is the manuscript presented in an intelligible fashion and written in standard English?

Reviewer #1: Yes

Reviewer #2: Yes

6. Review Comments to the Author

Reviewer #1: Thanks for the response to my comments. Pleasant to see that most of my questions has been solved.

1. The data sources are heterogeneous.

The Flesch-Kincaid Grade Level seems to be a proper measurement. Surprisingly, under this measurement, the three data sources are similar for the public to read. This evidence makes the conclusion sound.

2. The difference in the sizes of the three bodies of text.

Thanks for providing more information about the data. In the table 1, the author provided average document length which is consistent. The only question remains is that the total number of words are imbalanced. That's understandable since previous works shows that's acceptable.

For the other minor issues, the works revised them perfectly.

Reviewer #2: After reading the authors' response and the revised manuscript, most of my concerns have been addressed. However, I think it's better for the authors to add more detailed descriptions, such as some definitions, equations, or even simple examples in the Methods section to help the audience better understand your proposed methods.

7. PLOS authors have the option to publish the peer review history of their article (what does this mean?). If published, this will include your full peer review and any attached files.

Reviewer #1: No

Reviewer #2: No

---

## [Author Response · Author response to Decision Letter 1]

14 Nov 2021

Response to reviewers R2

Reviewer 1 and Reviewer 2 have answered all 5 questions with “Yes”, indicating approval of the manuscript.

Reviewer 1 

Thank you for your kind words. We are glad that our clarification of the heterogeneity of the sources of text, and the length of the documents meets with your approval. We have added an additional, very recent, reference that addresses computational text analysis of documents from different sources (Abdo et al. 2021).

Reviewer 2

Thank you for your approval of our revisions. 

We have now added specific examples of sentiment polarity scores that show the modification of adjectives (lines 172-180). The examples are taken from an authoritative source on the analytic method. 

We have also added formulae to provide more detail on the Kendall’s coefficient, which we used to test the similarity of the rank order of the topics in the three sources (lines 210-226).

In addition, we would note that two sections of text are highlighted because they were edited for clarity (Abstract lines 27-43, and Results lines 303-310). No substantive changes were made.

Thank you all for your care and consideration.

---

## [Decision Letter · Decision Letter 2]

10 Jan 2022

Narratives and Counter-Narratives in Religious Responses to Covid-19: A Computational Text Analysis

PONE-D-21-08822R2

Dear Dr. Idler,

We’re pleased to inform you that your manuscript has been judged scientifically suitable for publication and will be formally accepted for publication once it meets all outstanding technical requirements.

Kind regards,

Quanquan Gu

Section Editor

PLOS ONE

Reviewers' comments:

Reviewer's Responses to Questions

**Comments to the Author**

1. If the authors have adequately addressed your comments raised in a previous round of review and you feel that this manuscript is now acceptable for publication, you may indicate that here to bypass the “Comments to the Author” section, enter your conflict of interest statement in the “Confidential to Editor” section, and submit your "Accept" recommendation.

Reviewer #1: All comments have been addressed

Reviewer #2: All comments have been addressed

2. Is the manuscript technically sound, and do the data support the conclusions?

Reviewer #1: Yes

Reviewer #2: Yes

3. Has the statistical analysis been performed appropriately and rigorously? 

Reviewer #1: Yes

Reviewer #2: Yes

4. Have the authors made all data underlying the findings in their manuscript fully available?

Reviewer #1: Yes

Reviewer #2: Yes

5. Is the manuscript presented in an intelligible fashion and written in standard English?

Reviewer #1: Yes

Reviewer #2: Yes

6. Review Comments to the Author

Reviewer #1: Glad to see the revision in this work. Nice to see more details on specific examples of sentiment polarity scores and Kendall's coefficient. Also, glad to see the effort on explaining the heterogeneity of the sources of text.

Reviewer #2: After reading the author's response, all of my concerns have been addressed. Therefore, I would like to recommend the acceptance of the paper.

7. PLOS authors have the option to publish the peer review history of their article (what does this mean?). If published, this will include your full peer review and any attached files.

Reviewer #1: No

Reviewer #2: No

---

## [Editor Report · Acceptance letter]

27 Jan 2022

PONE-D-21-08822R2 

Narratives and Counter-Narratives in Religious Responses to Covid-19: A Computational Text Analysis 

Dear Dr. Idler:

I'm pleased to inform you that your manuscript has been deemed suitable for publication in PLOS ONE. Congratulations! Your manuscript is now with our production department. 

Kind regards, 

on behalf of

Dr. Quanquan Gu 

Section Editor

PLOS ONE